# Combination of Taurine and Black Pepper Extract as a Treatment for Cardiovascular and Coronary Artery Diseases

**DOI:** 10.3390/nu15112562

**Published:** 2023-05-30

**Authors:** Jordan Swiderski, Samy Sakkal, Vasso Apostolopoulos, Anthony Zulli, Laura Kate Gadanec

**Affiliations:** 1Institute for Health and Sport, Victoria University, Melbourne, VIC 3030, Australia; jordan.swiderski@live.vu.edu.au (J.S.); samy.sakkal@vu.edu.au (S.S.); vasso.apostolopoulos@vu.edu.au (V.A.); laura.gadanec@live.vu.edu.au (L.K.G.); 2Immunology Program, Australian Institute for Musculoskeletal Science, Melbourne, VIC 3021, Australia

**Keywords:** black pepper, cardiovascular diseases, taurine, terpenes

## Abstract

The shift in modern dietary regimens to “Western style” and sedentary lifestyles are believed to be partly responsible for the increase in the global burden of cardiovascular diseases. Natural products have been used throughout human history as treatments for a plethora of pathological conditions. Taurine and, more recently, black pepper have gained attention for their beneficial health effects while remaining non-toxic even when ingested in excess. Taurine, black pepper, and the major terpene constituents found in black pepper (i.e., β-caryophyllene; α-pinene; β-pinene; α-humulene; limonene; and sabinene) that are present in PhytoCann BP^®^ have been shown to have cardioprotective effects based on anti-inflammatory, antioxidative, anti-hypertensive and anti-atherosclerotic mechanisms. This comprehensive review of the literature focuses on determining whether the combination of taurine and black pepper extract is an effective natural treatment for reducing cardiovascular diseases risk factors (i.e., hypertension and hyperhomocysteinemia) and for driving anti-inflammatory, antioxidative and anti-atherosclerotic mechanisms to combat coronary artery disease, heart failure, myocardial infarction, and atherosclerotic disease.

## 1. Introduction

Natural products are well-known for their therapeutic value [1]. Throughout human history, they have played an important role in traditional medicine by treating a wide range of pathological ailments, offering effective alternatives to modern medicine [2]. Thanks to recent advances in scientific research and growing environmental awareness [3], there has been a significant increase in the use of natural and plant-based products to promote synergistic biological health benefits [4,5,6]. This is particularly important in western society where a shift in the modern dietary regimen towards a “Western diet” and an increase in sedentary lifestyles is believed to be, in part, responsible for the substantial increase in the prevalence of cardiovascular complications [7,8].

Cardiovascular diseases (CVDs) refer to a broad range of conditions affecting the heart and blood vessels [9]. They register a global annual death toll of more than 19.1 million and are the most prevalent cause of morbidity and mortality [10]. Alarmingly, the World Health Organization estimates that CVDs will contribute to over 22 million deaths in 2030 [11]. CVDs encompass a variety of diseases, including hypertension, atherosclerosis, coronary artery disease (CAD), heart failure (HF), myocardial infarction (MI) and stroke [9]. Atherosclerosis and hypertension are the leading causes of CVDs and are the most common underlying diatheses for most deaths [12]. Independent risk factors for CVDs include hypertension, hyperhomocysteinemia (HHcy), dyslipidemia, type-2 diabetes, and cholesterol [12,13,14]. The pathology of CVDs is complex, and oxidative stress, hyper-inflammation, endothelial dysfunction, and hyperlipidemia have been reported as common phenotypes in the pathogenesis of CVDs and have become the main targets of therapeutic intervention [15,16]. Conventional therapy commonly used to treat CVDs can be costly and may produce adverse side effects, unwanted drug–drug interactions and life-threatening toxicities [17], and thus, alternative approaches have been considered, including physical activity, as well as the use medicinal natural product supplementation, for CVD treatment [18,19,20]. 

Taurine, a natural product, is a non-toxic, semi-essential sulfur-containing amino acid (Figure 1A) that is abundant in human cardiac and vascular tissues [21,22]. Although synthesized endogenously from methionine and cysteine [23], the majority of taurine is obtained through dietary intake (e.g., seafood, meat, poultry and eggs) [21]. It is well-established that taurine has diverse physiological functions in humans and is seen to protect against several pathophysiological conditions [22,24,25]. Most of the effects mediated by taurine are reported in the cardiovascular system and include modulating ion transporters, membrane stabilization, anti-oxidation, anti-hyperlipidemia, anti-platelet, and anti-inflammatory activity, as well as modulation of blood pressure and vascular tone [22,24,25].

*Piper nigrum*, commonly known as black pepper, has been used to treat a variety of conditions [26]. The therapeutic value of black pepper is attributed to its constituents, terpenes or terpenoids, which are a diverse class of organic compounds present in essential oils obtained from plants [26,27]. Terpenes consist of over 30,000 compounds whose functional groups include alcohols, aldehydes, or ketones [28]. Abundant terpenes found in black pepper consist of *β*-caryophyllene (BCP) (Figure 1B), *α*-pinene (Figure 1C), *β*-pinene (Figure 1D), limonene (Figure 1E), *α*-humulene (Figure 1F), and sabinene (Figure 1G) [29]. The therapeutic merit of black pepper has been attributed to the ability of BCP to specifically activate cannabinoid 2 receptor (CB_2_R), resulting in downstream signaling pathways that mediate anti-inflammatory, anti-atherosclerotic and anti-oxidative properties, which have been correlated with beneficial cardiovascular effects [30]. Unlike BCP, studies involving *α*-pinene [31,32], *β*-pinene [31,32] and *α*-humulene [33] have demonstrated the inability of these terpenes to interact with CB_2_R. Several studies have noted the significant cardiovascular-protective effects of black pepper and its terpenes against a range of CVDs, including hypertension, atherosclerosis, CAD, and HF due to their anti-inflammatory, anti-oxidative, anti-hyperlipidemia and blood pressure-lowering abilities [34,35]. As such, black pepper provides novel therapeutic strategies to target CVD initiation and progression.

Herein, we raise the concept of a combination therapy of taurine and black pepper extract supplementation to provide synergistic protective effects against the development of CVDs through modulation of inflammation, oxidative stress, blood pressure, HHcy and key stages of atherosclerosis progression. 

## 2. Anti-Inflammatory Effects of Taurine and Terpenes

Acute and chronic inflammation play key roles in the pathogenesis, progression, and severity of CVDs [36]. Hypertension, atherosclerosis and CAD are characterized as chronic inflammatory diseases and are clinically associated with increased expression of important pro-inflammatory receptors (e.g., toll-like receptors (TLR) and receptor for advanced glycation end-products (RAGE)) and molecules (e.g., high-mobility group box-1 (HMGB-1), interleukin (IL)-1*β*, IL-6, tumor necrosis factor-alpha (TNF-*α*), monocyte chemoattractant protein-1 (MCP-1), nitric oxide (NO), prostaglandin E_2_ (PGE_2_), inducible nitric oxide synthase (iNOS), and cyclooxygenase (COX)-2) [8,37]. Thus, their downregulation and inhibition have been reported in the literature as potential therapeutic targets for CVDs [38]. Taurine and terpenes contain general anti-inflammatory effects (Table 1), which may be useful in combination therapy as due to their abilities to suppress the expression of pro-inflammatory cytokines and alleviate cell and tissue damage in multiple disease states [39,40,41,42]. The two natural compounds significantly mitigate the expression of pro-inflammatory mediators (e.g., IL-1*α*, IL-1*β*, IL-6, IL-13, IL-17, TNF-*α*, interferon-γ, MCP-1, and vascular endothelial growth factor) in order to reverse the severity of inflammation-mediated injury [42,43]. Administration of taurine (100 mM) in Wistar rats 30 days after coronary artery occlusion-induced ischemia significantly decreased myocardial infarct size and reduced expression of IL-6 and TNF-*α*, suggesting that taurine provides anti-inflammatory cardio-protective effects [44]. Understanding the anti-inflammatory targets of taurine and terpenes may provide further evidence of their therapeutic use for CVDs (Figure 2, Table 1), as well as other conditions such as inflammatory bowel disease [45], rheumatoid arthritis [45], and traumatic brain injury where inflammation is a well-known critical event [43].

### 2.1. Inflammatory Signaling Pathways

Nuclear factor kappa-light-chain-enhancer of activated B cells (NF-κB) is a central transcription regulator that plays a critical role in the activation of inflammatory responses by inducing transcription of pro-inflammatory genes [46]. During basal conditions, NF-κB is bound to the inhibitory protein nuclear factor of kappa light polypeptide gene enhancer in B-cells inhibitor alpha (IκB*α*) within the cytosol [47]. Upon stimulation, the inhibitor of NF-κB subunit *beta* (IκK*β*) phosphorylates and degrades IκB*α*, allowing NF-κB dimers p50 and p65 to translocate towards the nucleus where they upregulate the activation of pro-inflammatory genes [48]. The phosphorylation of IκB*α* is triggered by a diverse range of stimuli, including various ligands for IL-6 receptors, TNF receptor (TNFR), and pattern-recognition receptors (PRRs) (e.g., TLR and RAGE) [49,50,51], which can activate kinase-signaling transduction pathways consisting of p38-mitogen-activated protein kinase (MAPK), extracellular signal-regulated kinase (ERK) and Janus kinase (JAK) [52].

Taurine and terpenes both exert anti-inflammatory effects through inhibition of NF-κB and kinase-signaling pathways [53,54,55]. An increase in taurine metabolites, chloro-taurine (TauCl) and taurine-bromamine (TauBr) have been shown to modulate the myeloid-differentiation factor-88 (MyD88)-dependent TLR/NF-κB inflammatory signaling pathway in both human and mouse macrophages to disrupt the phosphorylation and degradation of IκB*α* [56,57], as well as prevent the activation and subsequent translocation of NF-κB [56,58]. Additionally, taurine can significantly reduce the expression signal transducer and activator of transcription 3 (STAT) 3 in the JAK-STAT signaling pathway [44], which is shown to play a vital role in cardio-protection, as inflammation associated with myocardial ischemia is mitigated through inhibition of JAK-STAT3-signaling pathways [44]. Black pepper terpenes have also been shown to attenuate pro-inflammatory cytokine expression through downregulation of kinase signaling pathways and inhibition of IκB*α* phosphorylation [44,59]. Limonene blocks the phosphorylation of IκB*α* and NF-κB to suppress lipopolysaccharide (LPS)-induced inflammation in acute lung injury [59]. α-pinene has been shown to inhibit IκK*β* activation and prevent NF-κB translocation [60]. In human chondrocytes, α-pinene (200 µg/mL) suppressed IL-1*β*-induced inflammation through suppression of NO, iNOS, matrix metallopeptidase (MMP)-1, and IL-13 pro-inflammatory mediators [61]. This anti-inflammatory effect was produced by the inhibition of c-Jun N-terminal kinase (JNK) phosphorylation. Additionally, α-pinene has also been shown to attenuate MAPK and NF-κB activation, resulting in the decreased expression of pro-inflammatory factors IL-6, TNF-*α*, NO, iNOS, and COX-2 [44]. The terpenes BCP, *α*-humulene, and *α*-pinene have been shown to inhibit the expression of NO synthase and COX-2 through inhibition of MAPK action through IKK-B inhibition [38], while limonene was shown to mitigate pro-inflammatory cytokines secretion as well as rescue iNOS, MMP-1, and MMP-3 gene transcription through inhibition of MAPK/JNK-dependent NF-κB activation pathways [62]. Pre-treatment with limonene (25, 50 and 75 mg/kg) inhibited release of pro-inflammatory cytokines (i.e., TNF-*α*, IL-1*β* and IL-6) and prevented phosphorylation of p38-MAPK, IκB*α*, NF-κB p65, JNK and ERK in mice with lipopolysaccharide (LPS)-induced acute lung injury [59]. Likewise, both α-pinene and *β*-pinene reduce the expression of genes associated with LPS-induced inflammation [63], suggesting that terpenes may mediate anti-inflammatory effects through regulation of kinase and NF-κB gene signaling. 

BCP has been identified as a CB_2_R agonist [64], which is primarily expressed in immune tissues and is known to modulate immune cell function to provide anti-inflammatory effects [65]. Exogenous administration of CB_2_R agonists is observed to inhibit inflammation by reducing the production of pro-inflammatory cytokines and attenuating oxidative stress in various inflammatory diseases [66,67]. BCP has been observed to inhibit the activation of NF-κB, thus attenuating the production of inflammatory mediators. Furthermore, this effect disappears with CB_2_R^−/−^ knockout expression, suggesting that BCP exerts its anti-inflammatory responses through activation of CB_2_R [68]. The cardioprotective effects of BCP on *isoproterenol* (ISO)-induced myocardial injury in Wister rats through activation of the CB_2_R has been observed. BCP modulated MAPK/ERK/JNK/NF-κB signaling to significantly suppress the expression of IL-1*β*, IL-6, TNF-α, iNOS, and COX-2 [69]. Additionally, this comprehensive study reported that BCP also attenuated mitochondrial dysfunction and improved the atherogenic index via a CB_2_R-independent reduction in dyslipidemia [69]. BCP has been shown to inhibit the expression of vascular cell adhesion molecule-1 [70], which is known to promote the adhesion of macrophages to the vascular endothelium in the development of atherogenic plaque [71]. These studies suggest that both taurine and terpene extracts present in black pepper may combine to exert effective anti-inflammatory responses and thereby provide beneficial cardioprotective effects in CVDs. 

The nucleotide-binding domain, leucine-rich containing family, pyrin domain-containing 3 (NLRP3) inflammasome is a critical component of the innate immune system and is responsible for the production of pro-inflammatory cytokines in response to cellular damage [72]. Recently, NLRP3 has been shown to play an indispensable role in the development of vascular diseases, including hypertension, atherosclerosis, and CAD, as well as diabetes and neurological disorders [73]. Increasing evidence supports NLRP3 as a new target to inhibit inflammatory diseases [74]. NLRP3 is regulated by NF-κB signaling pathways; therefore, taurine and terpenes have the ability to inhibit NF-κB as we have previously highlighted a pathway for preventing NLRP3 activation. Although taurine has been shown to attenuate in vitro and ex vivo inflammation through inhibition of NF-κB-dependent NLRP3 activation [75,76], studies on the isolated effect of black pepper terpenes on inflammasome inhibition are limited. A study conducted on the essential oil of pomelo peel, in which limonene (55.92%) was the most abundant compound, found that it significantly inhibited inflammation mediated by NLRP3 activation in rats [77]. Additionally, Makauy leaf ethanol extract consisting of limonene (5.6%) and BCP (4.9%) inhibited the in vitro expression of NLRP3 in J774A.1 mouse macrophage cells stimulated with LPS-induced inflammation [78]. Lastly, CB_2_R agonist JWH-133 has been shown to produce cardioprotective effects against MI in rats by suppressing NLPR3-related mechanisms, suggesting that BCP, a CB_2_R agonist, may also suppress inflammation through NLPR2 inhibition [79]. 

Collectively, these studies highlight the effects that taurine and black pepper terpenes have on inhibiting NF-κB, kinase- and inflammasome-signaling pathways and suppressing the expression of pro-inflammatory mediators. The potential combined use of taurine and terpenes may have synergistic effects in exerting effective anti-inflammatory responses, potentially providing effective treatment of CVDs. 

### 2.2. High-Mobility Group Box-1

HMGB-1 is an evolutionarily preserved, architectural, non-histone chromosome binding peptide that is found abundantly within the nucleus bound to DNA [80,81]. HMGB-1 facilities nuclear homeostasis by coordinating important nuclear functions, such as maintaining stability of the genome, acting as a chaperone for DNA, and regulating DNA transcription, replication, recombination, and repair [80,81]. Outside the nucleus, HMGB-1 drives cell stress responses by acting as a dynamic alarmin signal (which can also be referred to as a danger associated molecular pattern (DAMP)) that engages with PRRs of the immune system (e.g., TLR1/2/6, TLR4 and RAGE) to promote inflammation through the canonical NF-κB signaling pathway (discussed above) [81,82,83,84]. Elevated levels of HMGB-1 have been demonstrated in animal models of atherosclerosis, hypertension, CAD, MI, and HF [84]. Clinical studies have also reported increased HMGB-1 serum levels in patients with CAD [85,86], peripheral artery [87] and HHcy [88], suggesting that HMGB-1 has a critical pathogenic role during development and progression of CVDs. While the correlation between taurine and HMGB-1 has yet to be investigated in the literature, a study involving patients with bladder cancer noted elevated levels of taurine upregulate gene 1 (TUG1; long non-coding RNA transcripts that require taurine for gene upregulation) and the reversal of TUG1 knockdown by overexpression of HMGB-1 [89], thus suggesting a possible unknown mechanism between taurine, TUG1 and HMGB-1. The role that taurine and BCP have in attenuating HMGB-1-mediated inflammation and dysfunction remains elusive; however, there is evidence to suggest that there may be an unknown interlinking mechanism between taurine and HMGB-1 [90]. In mice with HMGB-1 deletion in intestinal epithelial cells, a diet high in fat, cholesterol and fructose increased taurine concentration in serum collected from portal blood samples when compared with wild type controls [90]. 

Moreover, BCP and CB_2_R activation may lower HMGB-1, contributing to a reduction in inflammation. CB_2_R knockout mice challenged with LPS to induce sepsis displayed reduced survival rate and significantly elevated serum IL-6, TNF-*α* and HMGB-1 levels when compared with CB_2_R expressing littermates [91]. Furthermore, treatment with GW405833 (10 mg/kg, a CB_2_R agonist) was able to significantly reduce serum levels of TNF-*α*, IL-6, and HMGB-1 6 h after LPS injection (5 mg/kg) [91]. Furthermore, activation of CB_2_R with GW405833 has been shown to induce intracellular HMGB-1 degradation via the autophagy-lysosome pathway in macrophages without modulating HMGB-1 mRNA expression [92], thus potentially preventing its release into the extracellular milieu where it can engage with immune system receptors and activate pro-inflammatory downstream signalling pathways. BCP’s anti-inflammatory properties may also be due to its ability to reduce HMGB-1 and its endogenous receptors as BCP was able to attenuate increased serum levels of HMGB-1 and suppressed protein expression of TLR4 and RAGE and their downstream inflammatory molecules (i.e., ERK, p38 and JNK phosphorylation, NF-κB, early growth response protein-1, and macrophage inflammatory protein-2 protein) in Kupffer cells from mice with galactosamine and LPS-induced hepatic injury [93]. 

Although the relationship between taurine, terpenes and HMGB1 remains elusive, these findings suggest that targeting HMGB-1-mediated inflammation through a combination of taurine and black pepper terpenes may be a potential mechanism to reduce inflammation associated with CVDs. 

### 2.3. Toll-like Receptors

TLRs are integral to immunity as they provide host surveillance by detecting molecular signatures present on pathogens (referred to as pathogen-associated molecular patterns) and launch mechanisms of inflammation to neutralize and eliminate infection [94]. Importantly, TLRs also provide protection against tissue injury in the absence of pathogenic infiltration by identifying DAMPs (e.g., HMGB-1) secreted by damaged, injured, and dying cells [94,95]. However, unregulated TLR activation results in a vicious cycle of unresolving inflammation and has been demonstrated to be fundamental in the development of CVDs. Thus, they have become a target to attenuate disease development [95,96]. Specifically, taurine has been shown to suppress inflammation via modulation of the TLR2 and TLR4 [97,98,99] signaling pathways. TLR2 and 4 are able to recognize bacterial cell wall components (e.g., lipopeptides, peptidoglycan and LPS), and stimulation of TLR4 causes receptor homodimerization of the MyD88- and Toll/IL-1 receptor domain-containing adaptor-inducing interferon-*β* (TRIF)-dependent signaling cascade, whereas TLR2 activation requires recruitment of TLR1 or 6 for normal receptor function via the MyD88-dependent pathway [100,101]. Male Sprague-Dawley rats challenged with LPS (25 mg/kg; TLR4 activating ligand) to induce acute lung injury showed that those which were pre-treated with taurine supplemented drinking water had decreased infiltration and activation of neutrophils and macrophages. Further, they lowered levels of TLR4, MyD88, NF-κB and p65, reduced pro-inflammatory cytokines (IL-6, IL-18) and increased anti-inflammatory cytokines (IL-4, IL-10) in serum and lung tissues of the treated rats [97]. Similarly, taurine provides hepatic protection by targeting the TLR4/MyD88/NF-κB pathway in rat models of chronic alcohol-fed [98] and thioacetamide-induced [99] liver injury, resulting in decreased lymphocyte infiltration and reduced inflammation (TLR4, MyD88, IκB, NF-κB, iNOS, IL-6, IL-1*β*, and TNF-*α*). Regarding TLR2, taurine has been shown to reduce neutrophil activation of MAPK signaling following *Streptococcus uberis* (TLR2 activating ligand) infection [102]. A supporting study also showed that pre-treatment of taurine in rat models with *Streptococcus uberis* mastitis had significantly decreased expression of iNOS and TNF-α, which was associated with the downregulation of TLR2 and NF-κB mRNA expression, as well as inhibition of the DNA binding activity of NF-κB [103].

Like taurine, terpenes may also exert anti-inflammatory effects by modulation of the TLR signaling cascade. In LPS activated RBL-2H3 cells (derived from a basophilic leukemia cell line), α-pinene, *β*-pinene and limonene displayed anti-inflammatory abilities by reducing the expression of genes associated with LPS-mediated inflammation [63]. Likewise, α-pinene attenuated activation of the MAPK and NF-κB pathways and reduced production of IL-6, TNF-α, NO, iNOS and COX-2 following LPS stimulated mouse intraperitoneal macrophages [104]. However, in human chondrocytes, α-pinene elicited a more potent inhibition of IL-1*β*-induced inflammation, NF-κB and JNK activation and expression of iNOS compared with *β*-pinene and limonene [61]. Importantly, BCP has been shown to ameliorate ISO-induced (85 mg/kg) MI through modulation of the heat shock protein-60/TLR2/4/MyD88/NF-κB pathway. This study reported that BCP treatment was able to markedly reduce infarct size, normalized electrocardiogram and blood pressure parameters and significantly reduced protein expression of MyD88, TLR2, TLR4, and TIR-domain-containing adaptor-inducing interferon-*β* (accessory molecule of TLR signal transduction) and levels of inflammatory markers (i.e., IL-1*β* and TNF-α) [105]. These findings suggest a direct relationship between the anti-inflammatory abilities of terpenes and TLR-mediated pathways involved in CVDs. 

These results suggest that combination therapy of taurine and black pepper terpenes may hold therapeutic cardiovascular-protective value for attenuating inflammation through modulation of TLR2 and 4 signaling pathways, which play a significant role in the development of CVDs. 

**Table 1 nutrients-15-02562-t001:** Summary of anti-inflammatory effects of taurine and terpenes in cell and animal models.

Compound	Dosage	Experimental Model	Outcome	Ref
α-humulene	50 mg/kg	In vivo LPS-induced inflammation in the paw of Wistar rats	↓ neutrophil migration, ↓ IL-5 ↓ NF-κB DNA binding ↓ IL-1*β* and TNF-α expression.	[106]
α-humulene	50 mg/kg, 22 days	In vivo female BALB/c mice	↓ Eosinophil recruitment ↓ NF-κB and activator protein-1 activation	[107]
α-pinene	50 mg/kg, 7 days	In vivo ISO-induced myocardial inflammation in Wistar albino rats	↓ cardiac injury biomarkers ↓ NF-κB signaling ↓ IL-6 and TNF-α expression	[108]
BCP	50–200 µg/mL	In vitro Mouse RAW267.4 macrophages	↓ ERK/p38-MAPK signaling ↓ COX-1 and COX-2	[109]
BCP	(0.2–25 µM)	In vitro LPS-induced inflammation in C57BL/6 mouse microglial cells	↓ IL-1*β*, TNF-*α*, PGE2, iNOS expression ↓ ROS inflammatory biomarkers	[110]
BCP	10 mg/kg	In vivo cisplatin-induced nephropathy in C57BL/6J mice	CB_2_R-dependant decrease in MCP-1, IL-1*β*, TNF-*α*, ICAM-1, neutrophil and macrophage infiltration	[111]
Limonene	50 mg/kg, 21 days	In vivo ISO-induced inflammation in male Wister rats	↓ MAPK/JNK/ERK/NF-κB signaling ↓ IL-1*β*, IL-6, and TNF-*α* expression	[112]
Limonene	20, 50, and 100 mg/kg	In vivo gastritis-induced male Sprague-Dawley rats	↓ NF-κB nuclear translocation ↓ intracellular Ca^2+^, IL-1*β*, IL-6, TNF	[113]
Sabinene	0.32–1.25 µL/mL, 1 h	In vitro LPS-induced inflammation in mouse Raw 264.7 leukemic macrophage cell line	Strong anti-inflammatory activity through potent NO scavenging and inhibition of iNOS	[114]
Taurine	100 mM, 30 days	In vivo myocardial ischemia-induced male albino Wister rats	↓ myocardial infarct size ↑ superoxide dismutase ↓ IL-6 and TNF-*α* expression	[44]
Taurine	3000 mg/day, 8 weeks	Clinical study of 50 patients with type-2 diabetes	↓ TNF-*α* ↑ superoxide dismutase ↑ catalase	[115]

*Abbreviations*: BCP, *β*-caryophyllene; Ca^2+^, calcium; CB_2_R, cannabinoid 2 receptor; COX, cyclooxygenase; ERK, extracellular signal-regulated kinase; ICAM-1, intercellular adhesion molecule 1; IκK*β*, nuclear factor kappa-*B* kinase subunit *beta*; IL, interleukin; iNOS, inducible nitric oxide synthase; JNK, c-Jun N-terminal kinase; LPS, lipopolysaccharide; MAPK, mitogen-activated protein kinase; MCP-1, monocyte chemoattractant protein-1; NF-κB, nuclear factor kappa-light chain enhancer of activated B cells; NO, nitric oxide; PGE2, prostaglandin 2; ROS, reactive oxygen species; TNF-*α*, tumor necrosis factor alpha; ↑, increase; ↓, decrease.

## 3. Anti-Oxidative Effects of Taurine and Terpenes

Oxidative stress is an important factor involved in the progression of CVDs. It is associated with upregulation of inflammation, endothelial dysfunction, and lipid peroxidation in various CVDs [116]. It involves the production of free radicals, oxidative reactive oxygenated species (ROS), with the capacity of oxidize lipids and proteins, cause DNA damage, injure cellular components, initiate apoptosis, induce inflammation, and cause cell death [116]. Various in vitro and in vivo experimental studies have highlighted the role that taurine and black pepper terpenes have in increasing cell viability, restoring mitochondrial function and improving antioxidant balance (Figure 3).

In contrast to taurine, terpenes have been shown to have a direct scavenging ability for free radicals which improves antioxidant status. The anti-oxidant activity BCP, *α*-pinene, *β*-pinene, and limonene have all been verified through 2,2-Diphenyl-1-picrylhydrazyl free radical scavenging assays [117,118,119], which is a widely used technique to assess the antioxidant activity of crude extracts from plants. The antioxidant effects of taurine are primarily attributed to its ability to suppress free radical production rather than acting as a traditional scavenger [120]. In cardiomyocytes and neuronal cells, taurine is observed to modulate intracellular calcium Ca^2+^ concentrations to protect against ROS-induced MI in rats and improve mitochondrial energy metabolism, respectively [22,121,122]. An increase in intracellular Ca^2+^ can impair mitochondrial membrane depolarization, causing a depletion in energy production. Additionally, increased ROS production during mitochondrial dysfunction is associated with decreased expression of mitochondrial ND5 and ND6 subunits of complex I of the electron transport chain (ETC), which can cause the ETC to slow, creating a bottleneck and increasing diversion of electrons to oxygen to form superoxide (O_2_^−^) free radicals [123]. Taurine is a component of mitochondrial tRNA [124]; therefore, it is suggested that the association between taurine and tRNA acts to regulate mitochondrial protein synthesis, thereby enhancing ECT activity, improving energy metabolism and protecting the mitochondria against excessive O_2_^−^ generation [120,125]. Additionally, taurine promotes expression of antioxidants such as superoxide dismutase (SOD), which is associated with inhibition of ROS-mediated inflammation [115]. In HK-2 cells, taurine has been shown to increase cell viability by inhibiting ROS production through increases SOD activity, decreased Ca^2+^ accumulation and restoration of mitochondrial function [126]. SOD is a protective antioxidant in CVDs [116]. Downregulation of SOD expression is shown to induce mitochondrial oxidative stress, promote cardiomyocyte hypertrophy [127], increase lipid peroxidation and progress the development of atherosclerotic plaque [128,129]. Concurrently, black pepper terpenes have also been shown to produce cell-protective antioxidant activity by increasing SOD, as well as additional antioxidants including glutathione peroxidase (GPx1) and catalase (CAT) [130,131,132], which are also key targets in the prevention of ROS-mediated CVDs [116]. Extensive research on the antioxidant effects of terpenes in essential oils (Table 2) has shown that these plant metabolites have potent antioxidant effects that could combine with taurine to protect cells against ROS-induced damage.

The free radical HOCl Is a major product of myeloperoxidase, an enzyme in atherosclerotic lesions that contributes to atherogenic lipoprotein dysfunction, reduced NO bioavailability, endothelial dysfunction, impaired vasoreactivity, and increased atherosclerotic plaque instability [133,134]. Our group has previously highlighted the role of taurine acting as an anti-oxidant and absorbing HOCl, but not scavenging it [24]. Taurine interacts with HOCl to produce a less reactive and toxic compound, TauCl, which is observed to preserve cellular function in response to oxidative stress [24,135]. Recent studies have suggested that TauCl may act as an anti-inflammatory modulator, inhibiting the production of pro-inflammatory factors [136]. In mice, administration of TauCl inhibits the activation of the pro-inflammatory transcription factors NF-κB and STAT3, which is associated with decreased expression of IL-6, TNF-α, and COX-2 and protects against experimentally induced colitis [137]. A study of LPS-induced pneumonia in high-fat fed mice observed that treatment with TauCl partly modulated attenuation of IL-6 and TNF-α expression associated with direct suppression of NF-κB [138]. Additionally, the anti-inflammatory effects of TauCl through inhibition of NF-κB are well documented in several experimental studies [139]. Evidence suggests that preventing HOCl formation by inhibiting myeloperoxidase may be a mechanism to prevent the progression of atherosclerosis. Mice treated with a myeloperoxidase inhibitor showed anti-atherogenic effects through alteration of inflammatory tone atherosclerotic lesions [140]. These results suggest that the anti-oxidative contributions of taurine to remove HOCl may contribute to the anti-atherosclerotic effect by promoting atherosclerotic lesion stabilization and preventing plaque rupture, while additionally, TauCl anti-inflammatory activity improves vascular health and prevents progression of atherosclerosis [141]. 

The ability of taurine and terpenes to improve antioxidant status may also be important in inhibiting the activation of ROS-dependent NF-κB expression associated with increases in pro-inflammatory signaling pathways [142,143]. Nuclear factor erythroid 2-related factor 2 (Nrf2) is a key transcription factor in the protection against oxidative stress and is responsible for regulating the redox balance in cells and tissue [144]. Nrf2 can regulate pro-inflammatory NF-κB signaling via multiple mechanisms that include inhibition of ROS-mediated NF-κB activation [145] and preventing the phosphorylation and subsequent degradation of IκK*β* to inhibit NF-κB activation [146]. Additionally, Nrf2 activation can also increase cellular oxygenase-1 (HO-1), which is a key enzyme that restores antioxidant homeostasis and has been linked to the prevention of vascular inflammation during CVDs and atherosclerosis through inhibition of IκK*β* degradation [147]. Taurine and terpenes present in black pepper have both been shown to protect against ROS-induced toxicity by promoting the translocation of Nrf2 and increasing the expression of HO-1 and antioxidants [148,149,150]. In porcine mammary epithelial cells, taurine pre-treatment increases cell viability against hydrogen peroxide (H_2_O_2_)-induced oxidative injury and enhances SOD expression dependent on Nrf2 activation [151]. Taurine is also capable of interacting with hypobromous acid, forming TauBr, which also activates Nrf2 and upregulates HO-1 [152,153,154]. In conjunction with taurine, α-pinene (5µg/mL) has been shown through radical scavenging assays to increase Nrf2-dependent activation of antioxidants SOD and GPx1, and CAT antioxidant gene targets [155], while BCP, a CB_2_R agonist, has also been observed to protect against ischemia-induced oxidative stress through upregulation of Nrf2 and HO-1 expression, as well as increased antioxidant activity of SOD and CAT, and GPx [156,157]. Hypercholesterolemia animal models show decreased expression of GPx [158]. Increased GPx expression is linked to the prevention of atherosclerosis [159]. Clinical studies show that GPx activity is inversely related to an increased atherosclerotic risk due to its ability to protect against ROS-mediated atherosclerotic lesion development [160,161].

Therefore, these studies highlight the potential dual synergistic antioxidant and oxidative-protective effects that combined treatment of taurine and black pepper terpenes may have in preventing oxidative damage in cardiovascular aetiologies and other oxidative diseases. The actions of taurine and terpenes may combine to both prevent free radical production and promote its removal from cells and tissues. 

**Table 2 nutrients-15-02562-t002:** Summary of antioxidant effects of taurine and terpenes in cell and animal models.

Compound	Dosage	Experimental Model	Outcome	Ref
α-pinene	10–400 µM, 24 h	In vitro H_2_O_2_-induced oxidative stress in U373-MG cell line	↓ H_2_O_2_-induced ROS production and decreased lipid peroxidation	[130]
BCP	200 mg/kg, 45 days	In vivo male albino Wistar rats	↑ SOD, CAT and GPx ↓ IL-6 and TNF-*α*	[162]
β-pinene	10 μM	In vitro Arsenic-induced oxidative stress in *O. sativa* seeds	↓ H_2_O_2_	[163]
Limonene	5–1000 μg/mL	In vitro H_2_O_2_ BALB/c mice lymphoid cells	↓ H_2_O_2_	[164]
Sabinene	0.08–0.16 μL/ml	In vitro RAW 264.7 murine macrophage cells	↑ N62O scavenging	[114]
Taurine	80 mM	In vitro ROS-induced oxidative stress in Rat H9c2 Cardiomyocyte cells	↑ Cell viability ↓ apoptosis ↓ intracellular Ca^2+^	[165]
Taurine	100 mg/kg/day, 10 days	In vivo tamoxifen-induced mitochondrial oxidative stress in Swiss albino rats	↓ mitochondrial lipid peroxidation ↓ O_2_^−^ ↑ mitochondrial antioxidants	[166]

*Abbreviations:* BCP, *β*-caryophyllene; CAT, catalase; Ca^2+^, calcium; GPx, glutathione peroxidase; HCl, hydrogen chloride; HOCl, hypochlorous acid; H_2_O_2_, hydrogen peroxide; NO, nitric oxide; O_2_^−^, superoxide; ROS, reactive oxygen species; SOD, superoxide dismutase; ↑, increase; ↓, decrease.

## 4. Anti-Hypertensive Effects of Taurine and Terpenes

Hypertension is clinically defined as abnormally elevated blood pressure and is one of the most common independent-predisposing factors for development of CVDs, including atherosclerosis, HF and CAD [167,168]. When unmanaged, elevated blood pressure can exert excessive mechanical stress upon vascular smooth muscle cells and triggers initiation of vascular remodeling events, such as endothelial dysfunction, impaired vasodilation, lumen narrowing, reduced vascular compliance, and thickening of artery walls, contributing to CVD progression [169,170]. Previous studies have investigated the anti-hypertensive and blood pressure-lowering abilities of taurine and terpenes (Table 3). Taurine oral supplementation has consistently been shown to reduce blood pressure in spontaneous hypertensive animal models [171,172] and hypertensive patients [173,174,175,176,177], and its deficiency has been correlated with the incidence of hypertension. In prehypertensive patients, a 12-week daily supplementation of taurine (1.6 g/day) was able to reduce mean SBP and DBP [173]. Strikingly, a high daily taurine dose (6 g/day) has been shown to significantly decrease MAP, SBP and DBP within 7 days, with no occurrence of organ toxicities [176,177]. This may in part be due to taurine having a direct, dose-dependent vasodilatory effect through modulating ion channels, including delayed outward potassium (K^+^) and large conductance Ca^2+^-activated K^+^ channels in the thoracic aorta of Wistar rats [178] and human radial arteries [179], respectively. Additionally, taurine may also promote relaxation and attenuate vascular constriction via non-specific endothelium-dependent mechanisms [180]. Recent clinical studies have demonstrated the ability of taurine to induce vasodilation in patients with endothelial dysfunction [181,182], which may be attributed to its ability to influence mechanisms of endothelium-dependent relaxation, such as upregulating endothelial nitric oxide synthase (eNOS) expression, increasing NO levels and bioavailability, and enhancing eNOS activity by augmenting phosphorylation at site Ser^1177^ [183,184,185,186]. Lastly, taurine may exert its anti-hypertensive ability by interacting with peptides of the renin–angiotensin system (RAS), an intrinsic hormonal system that plays a fundamental role in cardiovascular physiology and homeostasis by maintaining fluid and electrolyte balance, regulating vascular tone and resistance, and tightly controlling blood pressure [187,188]. The RAS consists of two counter-regulatory arms: the classical (e.g., angiotensin converting enzyme (ACE), angiotensin II (AngII) and angiotensin type I receptor) and alternative (e.g., ACE2, angiotensin 1–7 and angiotensin type 2 receptor) [188,189]. Cardiovascular pathologies, including hypertension, CAD, atherosclerosis, and HF, have been associated with upregulation of the classic arm and RAS dysfunction [188,189]. In neonatal rat cardiac cells, pre-treatment with taurine resulted in reduced responsiveness to AngII, lower levels of AngII-mediated protein synthesis and decreased degree of hyperplastic growth, and partially prevented intracellular Ca^2+^ increase after incubation with AngII (1 nM) to induce hypertrophy [190]. Moreover, in the hypothalamus, pituitary and adrenal glands of stress-induced hypertensive male Wistar rats, it was reported that taurine supplementation enhanced expression of ACE2 and inhibited gene and protein expression of ACE [191]. This suggests that taurine may promote RAS homeostasis by shifting from the deleterious classical arm to the alternative cardioprotective arm. 

Similarly, studies have also shown that terpenes in black pepper may exert blood pressure-lowering mechanisms through ion channels and endothelial- and vascular smooth muscle-dependent mechanisms. The protective properties of BCP in rats following isoproterenol-induced myocardial infarction was noted to reduce infarct size, decrease circulating levels of cardiac injury markers (e.g., myocardial creatine phosphokinase, creatine kinase-myocardial bound, lactate dehydrogenase, and cardiac troponin T) and normalize electrocardiograph traces and blood pressure (e.g., heart rate, systolic blood pressure (SBP), diastolic blood pressure (DBP) and mean arterial pressure (MAP)) [69,105]. The beneficial effect of BCP was noted to be via inhibition of the TLR4/MyD88/NF-κB signaling pathway and direct activation of CB_2_R [69,105]. Interestingly, in a clinical trial with smokers, it was shown that a 12-week BCP intervention (performed using a flavor capsule inserted into the cigarette filter) caused a reduction in brachial–ankle pulse wave velocity (an indicator of arterial sclerosis) but had no effect on blood pressure parameters (e.g., DBP, SBP, pulse pressure, heart rate and ankle brachial index) [192]. In male normotensive Wistar rats, an intravenous bolus dose of *α*-pinene or *β*-pinene caused transitory hypotension and tachycardia, which was hypothesized to be suggestive of a pronounced drop in vascular resistance and subsequent compensatory sympathetic peripheral vasoconstriction induced by activation of the baroreflex [193]. Similarly, a reduction in SBP via dominant sympathetic nervous system activation causing an overall “relaxed physiological state” has been observed in humans after smelling air containing low levels of *α*-pinene; however, this study reported that the highest concentration had a “stress-inducing effect”, and while no change in blood pressure was noted, there was an increase in pulse rate [194]. Furthermore, the relaxant effect of *α*-pinene and *β*-pinene may be via smooth muscle cell-dependent mechanisms, as both were reported to inhibit potassium chloride (KCl)-mediated contraction (80 nM) in rat ileums; however, the inhibitory effect of *β*-pinene was more pronounced [195]. This was also supported by a complimentary study that highlighted the antispasmodic and relaxing effect of *α*-pinene as it was able to inhibit contraction in response to KCl (60 mM) in guinea pig ileums after [196]. In addition, it is hypothesized that *β*-pinene invokes its anti-hypertensive ability through modulating ion channels. In mesenteric arterial rings from normotensive and *N*-nitro-L-arginine methyl ester (L-NAME; eNOS inhibitor)-induced hypertensive rats, *β*-pinene relaxed rings pre-contracted with KCl (80 mM), S(-)-BayK8644 (L-type Ca^2+^-channel activator), tetraethylammonium (non-selective K^+^ channel blocker) or phenylephrine (an alpha-1 adrenergic agonist) in the presence or absence of a functioning endothelium layer [197]. Moreover, *β*-pinene inhibited calcium chloride- and sodium-orthovanadate-mediated contraction, suggesting that *β*-pinene induces endothelium-independent vasorelaxation by inhibiting Ca^2+^ influx and decreasing Ca^2+^ sensitivity through L-type Ca^2+^ channels [197]. A supporting study also demonstrated the ability of *β*-pinene to inhibit rat uterine contraction by acting as a weak L-type Ca^2+^ channel activator and interrupting the influx of Ca^2+^ [198]. Conversely, there is contradictory evidence regarding the potential beneficial vasoactive effects of limonene. In ISO (85 mg/kg) challenged Wistar rats, pre-treatment with limonene, prevented increases in blood pressure parameters (i.e., SBP, DBP and mean arterial pressure) [112]. Similarly, limonene-treated Wistar rats fed an 8-week high-fat diet (42.2% beef tallow) with and without L-NAME-treated drinking water (80 mg/L) had significantly reduced blood pressure; however, in the control group rats fed a normal chow diet, limonene did not alter blood pressure [199]. Additionally, limonene was able to reduce circulating levels of lipids (e.g., total cholesterol (TC), total triglycerides (TG), free fatty acids, and phospholipids) and normalized enzymatic activity of superoxide dismutase, catalase, glutathione peroxidase and reductase, highlighting limonene as having both lipid- and blood pressure-lowering properties [199]. In isolated Wistar rat aortas, endothelium denudation significantly altered relaxation and constriction induced by limonene pre-contracted with K^+^ (60 mM), perillyl alcohol or phorbol 12,13-dibutyrate (1 µM; a protein kinase C activator) [200]. Moreover, the relaxing effect of limonene was partially blocked by L-type Ca^2+^ channel agonist Bayk8644 in the thoracic aorta of male Wister rats [200]. Additionally, limonene was able to attenuate isoproterenol-mediated myocardial infarction in Wistar rats treated with limonene by preventing intracellular Ca^2+^ increases in cardiomyocytes and decreasing superoxide dismutase and Ca^2+^-induced mitochondrial oxidative stress [201]. However, in isolated rat myometria, low concentrations of limonene caused dose-dependent contractions, which was reduced by nifedipine (10^−8^ M; L-type Ca^2+^ channel blocker), tetraethyl-ammonium (10^−3^ M) or theophylline (10^−5^ M), while paxilline (10^−5^ M) reversed this and caused relaxation [202]. This study also reported that limonene reduced cyclic adenosine monophosphate synthesis, activated A_2A_ receptors and mediated voltage-gated Ca^2+^ channel opening (the major mechanisms utilized in the myometrium to induce smooth muscle contraction) [202]. Furthermore, in human epithelial cells, limonene has been shown to decrease the mRNA of ACE2 and potently inhibited ACE2 activity and protein expression [203]. Finally, there is also evidence that humulene may have potential vasoactive abilities as it demonstrated a concentration-dependent reduction and inhibitory effect on histamine production in human leukemic mast cells stimulated with compound 48/80 [113]. Moreover, humulene was able to decrease intracellular Ca^2+^ concentration and mobilization and increased cyclic adenosine monophosphate levels in a dose-dependent manner [113]. 

The studies highlighted above suggest that combined taurine and black pepper terpenes supplementation may promote blood pressure-lowering effects by promoting vasodilation, endothelial- and vascular smooth muscle-dependent relaxation and regulation of the RAS to promote anti-hypertensive mechanisms and provide cardiovascular protective effects. 

**Table 3 nutrients-15-02562-t003:** Summary of the hypotensive and blood pressure-lowering abilities of taurine and terpenes.

Compound	Dosage	Experimental Model	Outcome	Ref
α-pinene	50 and 100 mg/kg I. V	In vivo ISO-induced myocardial infarction in male Wistar rats	↓ SBP, DBP and heart rate	[113]
*Citrus aurantium* (9.6% *ß*-pinene and 8.54% limonene)	0.05–0.2%	Ex vivo isolated thoracic aorta of C57BL/6 mice	↑ vasorelaxation and ↓ Ca^2+^ influx	[204]
*Lamiaceae* (α-pinene and BCP)	In vivo, 5,10, 20, and 40 mg/kg I.V. Ex vivo, 1–1000µg/mL	In vivo male Wistar rats and isolated mesenteric artery	↓ blood pressure, ↑ vasorelaxation and caused tachycardia	[205]
Limonene	(0.01, 0.1, and 0.01% *v*/*v*)	Ex vivo isolated thoracic aorta of C57BL/6 mice	↑ vasorelaxation and ↓ Ca^2+^ influx	[206]
Taurine	10 µM–10 mM	Ex vivo isolated human radial artery	↑ vasorelaxation and ↓ Ca^2+^ influx	[179]
Taurine	1–2% (*w*/*v*) ad libitum, 3 weeks	In vivo L-NAME-induced hypertension in male Sprague-Dawley rats	↑ endothelial NO and ↓ blood pressure, AngI and AngII	[207]
Taurine	2.5%, ad libitum	In vivo male Wistar rats	↓ SBP, DBP, and mean arterial pressure, and ↑ SOD and eNOS	[208]

*Abbreviations:* AngI, angiotensin I; AngII, angiotensin II; BCP, *β*-caryophyllene; Ca^2+^, calcium; DBP, diastolic blood pressure; eNOS, endothelial nitric oxide synthase; NO, nitric oxide; SBD, systolic blood pressure; SOD, superoxide dismutase; ↑, increase; ↓, decrease.

## 5. Abilities of Taurine to Modulate HHcy Pathology

An elevated plasma level of the amino acid homocysteine (Hcy) is an independent risk factor for the development of CVDs [209]. Hcy is a sulfur-containing amino acid that is biosynthesized as a byproduct of methionine metabolism, and with the assistance of vitamin B_6_, B_9_ and B_12_, can be reconverted into methionine or cysteine [210]. Within westernized populations, HHcy has been linked to a dietary intake of methionine-rich foods (e.g., eggs, dairy products, meat, and poultry), deficiencies in B vitamins, some prescription medications and pre-existing comorbidities [211]. Incredibly, a 5 µmol/L deviation from normal Hcy levels has been correlated with a 20% increased risk of CVD development [209]. Moreover, as there has been a significant shift in modern dietary regimens towards a “Westernized diet”, it is estimated that 30% of the general population have some severity of HHcy and that HHcy is present in up to 41% of patients presenting with atherosclerosis [212]. Importantly, a recent population-based, cross-sectional study reported that Hcy levels significantly increase after the age of 50 years and are markedly elevated in males compared with females, suggesting that Hcy levels may contribute to the gender differences that are associated with atherosclerosis and CAD [213]. An elevated Hcy concentration has consistently been shown to cause endothelial and vascular dysfunction, promote aggressive oxidative stress and inflammation, and drive pro-atherosclerotic mechanisms [214,215,216]. Current therapeutic efforts in clinical trials to treat HHcy have focused on replenishing vitamin B deficiencies or supplementing with B vitamins; however, patient outcomes have been disappointing, and no reductions in CVD risk have been reported [217,218]. On the other hand, abnormally low basal levels of Hcy have been correlated with detrimental consequences, such as increased risk of Alzheimer’s disease development [219], idiopathic peripheral neuropathy [220] and increased susceptibility to oxidative stress by impairing the synthesis of glutathione [221]. Due to the complex and multifaceted mechanisms ubiquitous to HHcy and CVDs, the ability of taurine and black pepper extract to antagonize them renders a combination therapy an appealing treatment option. 

We previously showed that taurine supplementation (2.5%) is able to normalize hyperhomocysteinemia and reduces atheroma in the left coronary artery of male New Zealand White rabbits fed a high-methionine and high-cholesterol diet by reducing endothelial cell endoplasmic reticulum stress and apoptosis [222]. Importantly, the ability of taurine to markedly reduce Hcy plasma levels has been replicated in middle-aged Korean women enrolled in a 4-week taurine supplementation (3 g/day) study [223]. However, conflicting evidence from an animal study involving rats fed a 4-week choline-deficient diet. It was determined that taurine (2.5%) was unable to prevent plasma Hcy increase but was able to normalize beta-homocysteine S-methyltransferase and choline dehydrogenase (genes involved in the re-methylation pathway) and reduced the expression of genes important for regulating phosphatidylcholine synthesis and fat accumulation in the liver [224]. The anti-oxidant effect of taurine may be important during HHcy as taurine has been reported to oppose the oxidative effect of Hcy in isolated rat myocardial mitochondria [225]. This study showed that Hcy inhibited mitochondrial uptake of ^45^Ca and Ca^2+^-ATPase activity and stimulated the production of H_2_O_2_ and O_2_^−^ in a dose-dependent manner, while taurine had the opposite effect [225]. Additionally, Hcy has been shown to directly initiate endothelium reticulum stress in endothelial cells of blood vessels by triggering a protein unfolding response and activation of glucose-regulated protein 78 (GRP78) [216,226]. We and others have shown the ability of taurine to reduce the induction of GRP78 and extracellular SOD mRNA expression [216,226]. As GRP78 has been identified as a danger-associated molecular pattern for TLR2 [227], taurine may reduce inflammation caused by Hcy through modulating TLR pathways. Hcy has been correlated with endothelial cell injury via upregulation of TLR4/NF-κB signaling [228], resulting in impaired cell viability, DNA methylation, decreased SOD activity and increased expression of reactive oxygen species (i.e., H_2_O_2_, and oxLDL) and lectin-type oxLDL receptor-1 (LOX-1) [228]. Moreover, Hcy and LPS appear to have synergetic effects, acting in unison to differentiate macrophages into the pro-inflammatory subtype, potentially facilitating the induction of atherosclerotic pathology [229]. These findings suggest possible interlinking mechanisms between taurine, Hcy and TLRs, which warrant further exploration. 

## 6. Anti-Atherosclerotic Effects of Taurine and Terpenes

Atherosclerosis is a chronic inflammatory disease characterized by the aggregation of plaque on the endothelial lining of arteries [230]. Over time, this accumulation of plaque is prone to rupturing, which is associated with 75% of acute coronary complications, contributing to CAD, peripheral artery disease, HF, stoke, and/or death [231]. Thus, attenuating plaque angiogenesis is an important process in preventing the development of atherosclerosis [232]. Early-stage plaque formation is initiated by dysfunction of the endothelial cell lining of arteries in lesion-prone vasculature as a result of increased inflammation, oxidative injury, reduced vascular tone, hypercholesterolemia, and hyperglycemia [233]. Endothelial dysfunction is accompanied by the internalization of low-density lipoprotein (LDL) within the intima caused by an enhanced inflammatory response [234]. Here, LDL is susceptible to oxidative modification via ROS to produce oxidized LDL (oxLDL) [235]. oxLDL can then be absorbed by macrophages to promote foam cell formation where they contribute to the formation of atherogenic plaque [235,236] (Figure 4). The osmoregulatory, anti-inflammatory, and antioxidative effects of taurine and terpenes that we previously discussed may contribute to the mechanism responsible for their anti-atherogenic effects. Additionally, the cholesterol-lowering effects of these natural products, as highlighted in Table 4, may also contribute to their preventative role in the initiation and development of atherosclerosis.

Taurine and terpenes have both independently been shown to prevent the development of atherosclerosis in various animal models, as highlighted by the studies listed in Table 4, which indicates that their combined supplementation may enhance antiatherogenic efficacy. Epidemiological studies have shown that LDL is an independent risk factor for atherosclerosis and CVDs [237,238]. It is evident that taurine and terpene supplementation prevents hypercholesterolemia by reducing the accumulation of plasma lipids, LDL, and very low-density lipoprotein (VLDL) in high-fat/high-cholesterol fed animals [233,239,240,241,242]. Clinically, hyperlipidemia is primarily treated via antihyperlipidemic drugs, which are known to be poorly tolerated in humans and are associated with several adverse effects [243]. Therefore, combining the effects of natural products such taurine and black pepper terpenes may be more advantageous than current antihyperlipidemic medication for reducing the concentration of plasma lipids and preventing atherogenic development. Taurine (2% (*w*/*w*)) has been shown to reduce aortic lesions and lipid accumulation in apolipoprotein E-deficient mice (ApoE^−/−^), a well-established model for studying atherosclerosis [244]. Furthermore, taurine (1% in drinking water) has been shown to attenuate endothelial dysfunction by modulating the decrease in LOX-1 expression, an endothelin receptor responsible for the internalization of oxLDL [183]. The same study also reported that taurine significantly attenuated overexpression of intracellular adhesion molecule (ICAM-1), which is an important marker of endothelial dysfunction [183]. These results suggest that taurine may be able to attenuate atherogenic plaque accumulation by preventing foam cell formation through a reduction in LOX-1. Additionally, terpenes, such as lactones, are also reported to inhibit oxLDL-induced foam cell formation due to a reduction in LOX-1 expression in ApoE^−/−^ high-fat-fed mice [245]; however, further studies are required to investigate the ability of terpenes present in black pepper to inhibit LOX-1 expression. 

Inflammation plays a key role in the progression of atherosclerosis. Formation of foam cells and oxidative damage to endothelial cells can result in increased secretion of pro-inflammatory cytokines (IL-1*β*, IL-6, IL-8, COX-2, and TNF-α) to amplify the inflammatory response in arterial lesion formation [246]. MMP, and growth factors can stimulate lesion growth, while monocyte adhesion molecules, vascular cell adhesion protein 1 (VCAM-1), ICAM-1, and MCP-1, can stimulate macrophage adhesion to the intima [247]. Moreover, MMP-2, and MMP-9 can weaken the fibrous cap, increasing the risk of plaque rupture [248]. Taurine and terpenes have been shown to reduce oxidative injury and downregulate this inflammatory cytokine response through inhibition of NF-κB-signaling to potentially prevent the development of atherosclerosis. 

BCP supplementation is associated with potent hypolipidemic and antioxidant effects to reduce the risk of atherogenic and CAD in hypercholesterolemia-induced rats [249,250]. BCP is a CB_2_R agonist. Studies investigating the role of the cannabinoid system in regulating oxLDL-induced inflammation and foam cell formation in mouse and human macrophages have reported that activation of the CB_2_R is associated with a decreased expression of inflammatory cytokines through inhibition of MAPK/ERK1/2/NF-κB-signaling [251], as well as prevention of the intracellular accumulation of oxLDL in macrophages [252]. These anti-atherogenic and anti-inflammatory effects were reversed by the presence of selective CB_2_R antagonist SR44628, thus demonstrating the protective role of CB_2_R activation in regulating foam cell formation and inflammation as a potential target for treating atherosclerosis through BCP. Activation of CB_2_R reduced plaque area in the aortic root of ApoE^−/−^ mice that was associated with downregulation of adhesion molecules ICAM-1 and VCAM-1, along with macrophage infiltration [253]. These results are supported by additional studies on CB_2_R agonists, in which activation of the CB_2_R resulted in reduced atherosclerotic plaque size in the aortic root of ApoE^-/-^ mice via inhibition oxLDL-induced NF-κB cytokine gene expression [254]. Limonene has also been reported to contain anti-hyperlipidemic properties, such as reducing plasma lipid content, fasting glucose levels, and LDL in high-fat-fed obese mice [255]. These effects were associated with activation of peroxisome proliferator-activated receptor-α (PPARα), a nuclear transcription factor for the regulation of lipid metabolism in the liver [256]. Previous studies have demonstrated the critical role of PPARα in ameliorating hyperlipidemia through lipoprotein metabolism [257], which suggests that limonene may protect against dyslipidemia through regulation of PPARα signaling. 

Myeloperoxidase derived from monocytes and macrophages produces HOCl, which can induce tissue damage and is implicated in the pathogenesis of endothelial dysfunction [258]. Similar to oxLDL, HOCl-modified LDL is also readily taken up by macrophages to stimulate foam cell production and arterial plaque accumulation [259]. As discussed, taurine reacts with ^-^OCl to produce TauCl, a less reactive substrate. Therefore, neutralization of HOCL by taurine may suppress plaque accumulation by inhibiting HOCl-modified LDL uptake. Additionally, TauCl can inhibit NF-κB to suppress the activation of the pro-inflammatory mediators TNF-α, PGE_2_, MCP-1 in activated macrophages [260]. These findings suggest that taurine and terpenes may exert anti-atherogenic effects through multiple mechanisms of attenuating plaque atherogenesis, suggesting their potential dual synergistic cardio-protective effects. 

Furthermore, imbalances in the gut microbiome have been linked to the pathogenesis of atherosclerosis [261,262]. Short-chain fatty acids produced by gut bacterial fermentation have a beneficial effect against systemic inflammation, inhibiting pro-inflammatory cytokines and promoting the preservation of endothelial function, thereby exhibiting anti-atherosclerotic actions [263,264]. Imbalances in gut microbiota, resulting in pathological bacteria can decrease the availability of short-chain fatty acids and induce system inflammation to aggravate plaque development in atherosclerosis [265,266]. Taurine and terpenes are both known to contain antimicrobial activities [267]. In healthy mice, taurine treatment (165 mg/kg) inhibits the growth of harmful bacteria and increases the production of short-chain fatty acids [268]. Major black pepper terpene limonene has been shown to positively modulate gut microbiota in high-fat-fed animal models [269], while a meta-analysis on the antibacterial actions of α-pinene reported that the terpenoid provides protective effects against a wide variety of pathogenic gut bacterial, including *E. coli* [270]. These studies suggest that one possible aspect of taurine and terpenes anti-atherosclerotic actions may be the modulation of the gut microbiome and potential increases in short-chain fatty acids. 

Overall, these studies demonstrate that taurine and terpenes constituents in black pepper can provide anti-atherogenic effects through inhibiting plaque angiogenesis and reducing inflammation. Additionally, these natural compounds show multifaceted abilities to reduce hypercholesterolemia and hyperlipidemia, inhibit lipid oxidization, and potentially modulate gut microbiota to suppress systemic inflammation. Their combined supplementation may show promise in preventing multiple pathways for the development and progression of atherosclerosis.

**Table 4 nutrients-15-02562-t004:** Summary of anti-atherosclerosis effects of taurine and terpenes.

Compound	Dosage	Experimental Model	Outcome	Ref
α-pinene	25, 50, and 100 mg/kg, 7 days	In vivo alloxan-induced diabetes in male Wistar rats	↓ plasma, TC, TGA VLDL, and LDL	[271]
*β*-pinene	25, 50, and 100 mg/kg, 7 days	In vivo alloxan-induced hyperlipidemia in male Wister rats	↓ plasma TGA, VLDL, and LDL levels	[272]
BCP	1 mL/kg, 3 days	In vivo Triton WR-1339-induced hypercholesterolemia in female Wistar rats	↓ cardiac TC and TG levels ↓ atherogenic and coronary risk index ↓ ROS	[158]
BCP	100 mg/kg/day, 2 days	In vivo ISO-induced myocardial infarction in rats	↓ IL-1*β*, IL-6, iNOS, COX-2 and TNF-α expression ↓ HMGB-1 expression	[69]
Taurine	0.5–10 g/kg, 2 weeks	In vivo high-cholesterol-fed male rats	↓ plasma TC, TG, LDL, and hepatic TG levels	[273]
Taurine	0.3% (*w*/*v*) ad libitum, 24 weeks	In vivo Watanabe heritable hyperlipidemic rabbits	↓ aortic lesions and cholesterol ester in arteries and macrophage migration	[274]
Taurine	2.5% to diet ad libitum, 4 weeks	Male New Zealand White rabbits on high-cholesterol and Hcy diet	↓ endothelial cell apoptosis and left main coronary artery atherosclerosis.	[222]
Taurine	1% (*w*/*v*) ad libitum, 14 days	Buthionine sulfoximine-induced oxidative stress in New Zealand white rabbits	↓ blood pressure, plasma ROS and LOX-1 expression	[275]

*Abbreviations:* ApoE^-/-^, apolipoprotein; BCP, *β*-caryophyllene; CAT, catalase; COX, cyclooxygenase; eNOS, endothelial nitric oxide; GPx, glutathione peroxidase; HMGB-1, high-mobility group box-1; HDL, high-density lipoprotein; IL, interleukin; iNOS, inducible nitric oxide; ISO, *isoproterenol*; *LDL*, *low-density lipoprotein*; *LOX-1*, lectin-type oxLDL receptor-1; oxLDL, oxidized low-density lipoprotein; ROS, reactive oxygen species; SOD, superoxide dismutase; TC, total cholesterol; TG, total triglycerides; VLDL, very low-density lipoprotein; ↓, decrease.

## 7. Conclusions

This comprehensive literature review provides a plethora of evidence supporting the use of taurine and the terpene constituents (i.e., β-caryophyllene, α-pinene, β-pinene, α-humulene, limonene, and sabinene) found in PhytoCann BP^®^ black pepper extract [276] to target risk factors for CVDs (i.e., hypertension and hyperhomocysteinemia). These risk factors are targeted in the following ways: promotion of anti-inflammatory effects through suppression of TLR/NF-κB/MAPK-dependent proinflammatory signaling pathways, increased anti-oxidative effects by restoring mitochondrial health, prevention of the production of oxidative free radicals, and promotion of scavenging of free radicals. The anti-atherogenic effect is associated with the suppression of hypercholesterolemia and hyperlipidemia, and the inhibition of cellular adhesion molecules, which prevents atherogenic plaque development. These mechanisms of taurine and terpenes could potentially combine to act in synergy in novel treatment for CVDs, including CAD, HF, MI, and atherosclerotic disease. Importantly, we also show the unique cardioprotective abilities of taurine and black pepper extract. For example, taurine appears to be able to target RAS, while black pepper extract is able to quench HMGB-1 release, thus providing the theoretical basis for the development of a combined treatment. Based on previous dosages used in the literature and clinical studies, we propose a theoretical dose of 500 mg–3 g of taurine [115,173,176,177,223] and 5–150 mg of black pepper extract [69,105,162] to be taken orally in capsule form, twice a day. Further clinical studies are required to confirm this hypothesis that a combined taurine and black pepper supplementation may be a viable adjunct treatment of CVD and its complications in humans.

## Figures and Tables

**Figure 1 nutrients-15-02562-f001:**
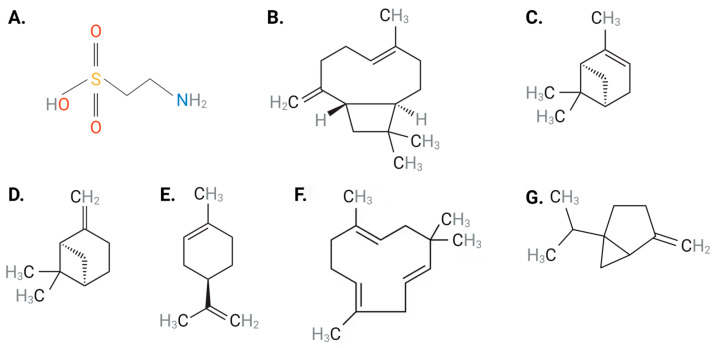
Chemical structures of (**A**) taurine, (**B**) BCP, (**C**) *α*-pinene, (**D**) *β*-pinene, (**E**) limonene, (**F**) *α*-humulene and (**G**) sabinene. *Abbreviations:* BCP, *β*-caryophyllene. Black: carbon, gray: hydrogen: blue: nitrogen; red: oxygen, yellow: sulfur Figure designed with Biorender.com.

**Figure 2 nutrients-15-02562-f002:**
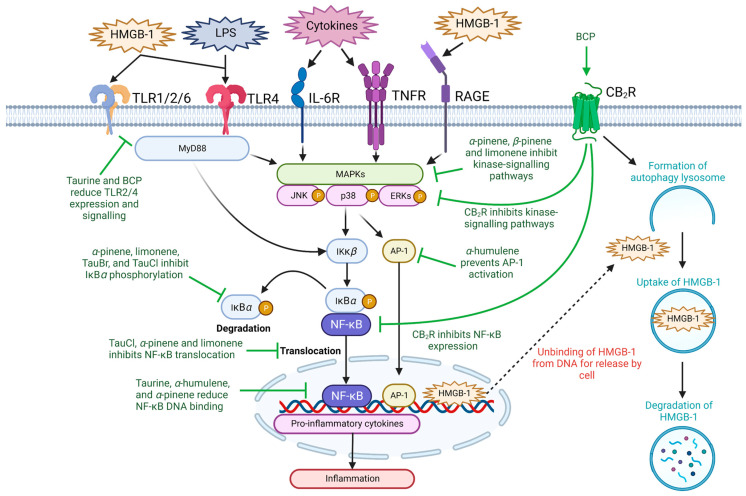
The anti-inflammatory abilities of taurine and terpenes through modulating NF-κB signaling. Taurine, its metabolites, and the major terpenes present in black pepper may suppress inflammation through preventing NF-κB-mediated proinflammatory cytokine production. The current literature suggests that this occurs through inhibition of upstream kinase signaling pathways, such as receptors of the immune system (i.e., TLR, RAGE, TNFR and IL-6R), MAPK/p38/ERK/JAK-signaling, release of HMGB-1, NF-κB activation, translocation or reducing NF-κB DNA binding ability. Ɑ-humulene may also inhibit proinflammatory cytokine production through downregulation of AP-1 signaling. *Abbreviations:* AP-1, activator protein-1; BCP, *β*-caryophyllene; CB_2_R, cannabinoid type 2 receptor; ERK, extracellular signal-regulated kinase; HMGB-1, high-mobility group box-1; IκB*α*, nuclear factor of kappa light polypeptide gene enhancer in B-cells inhibitor alpha; IκK*β*, nuclear factor kappa-*B* kinase subunit *beta*; IL-6R, interleukin 6 receptor; JNK, c-Jun N-terminal kinase; LPS, lipopolysaccharide; MAPK, mitogen-activated protein kinase; MyD88, myeloid differentiation protein 88; NF-κB, nuclear factor kappa-light-chain-enhancer of activated B cells; P, phosphorylation; RAGE, receptor for advanced glycation end products; TLR, toll-like receptor; TNFR, tumor necrosis factor receptor. Figure designed with Biorender.com.

**Figure 3 nutrients-15-02562-f003:**
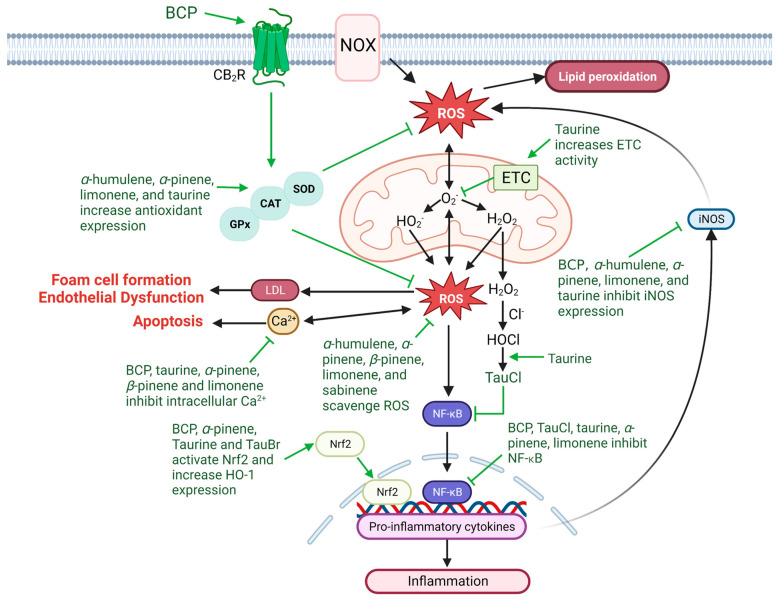
The antioxidative abilities of taurine and terpenes. The mitochondria, NOX, and iNOS are known to contribute to the intracellular accumulation of ROS, such as O_2_^−^, H_2_O_2_, HO_2_^−^, and HOCl. ROS can induce lipid peroxidation, mitochondrial ROS production, and the oxidation of LDL to contribute to foam cell formation and endothelial dysfunction in atherosclerosis. Furthermore, ROS can lead to an increase in intracellular Ca^2+^ which induces apoptotic pathways, increases the production of ROS through NF-κB/iNOS pathways, and contributes to inflammation through the increased production of proinflammatory cytokines via NF-κB-dependent signaling pathways. Taurine and the major terpenes present in black pepper both possess antioxidant activity. While terpenes act as radical scavengers, taurine exerts antioxidant activity by maintaining mitochondrial health through regulation of ETC activity. Taurine can also interact with ^-^OCl to form TauCl, which similar to terpenes, can inhibit proinflammatory responses and the expression of ROS-generating cytokine iNOS by suppressing NF-κB activation. Furthermore, taurine and terpenes enhance antioxidant activity (SOD, CAT, and GPx) while preventing the intracellular accumulation of Ca^2+^. The major terpene in black pepper, BCP, exerts its antioxidant effects through activation of CB_2_R, and both BCP and taurine can increase the expression of antioxidant HO-1 through activation of the antioxidant gene transcription regulator Nrf2. *Abbreviations:* BCP, *β*-caryophyllene; Ca^2+^, calcium; CAT, catalase; CB_2_R, cannabinoid type 2 receptor; ETC, electron transport chain; GPx, glutathione peroxidase; HO_2_^−^, hydroperoxyl radical; H_2_O_2_, hydrogen peroxide; HOCl, hypochlorous acid; HO-1, heme oxygenase-1; iNOS, inducible nitric oxide synthase; LDL, low-density lipoprotein; NF-κB, nuclear factor kappa-light-chain-enhancer of activated B cells; NOX, nicotinamide adenine dinucleotide phosphate oxidase; Nrf2, nuclear factor erythroid 2-related factor 2; O_2_^−^, superoxide; ROS, reactive oxygen species SOD, superoxide dismutase; TauCl, chloro-taurine. Figure designed with Biorender.com.

**Figure 4 nutrients-15-02562-f004:**
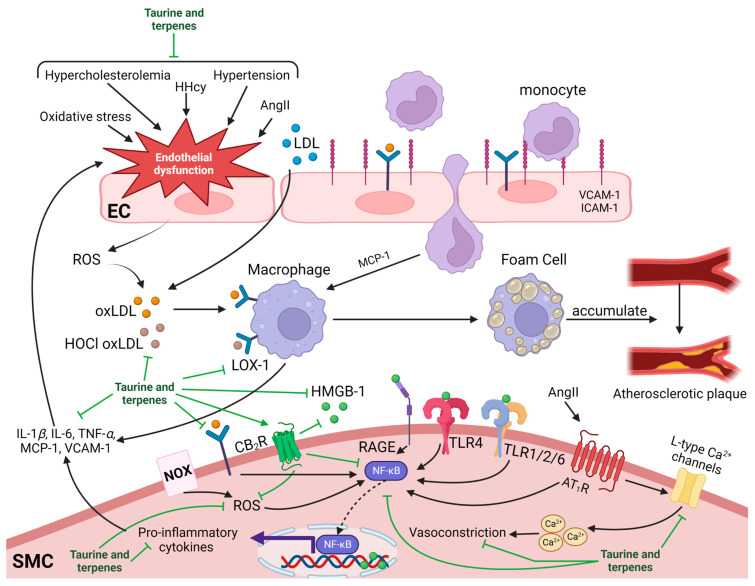
The protective effects of taurine and terpenes in preventing the progression of atherosclerosis. Inflammation, oxidative stress, hypercholesterolemia, hypertension and AngII can all contribute to endothelial dysfunction that is accompanied by the infiltration of LDL into the intima of arteries. LDL is modified by ROS to produce oxLDL. Inflammation promotes the recruitment and differentiation of monocytes into macrophages via the upregulated expression of the adhesion molecules ICAM-1, VCAM-1, and MCP-1. Macrophages absorb oxLDL to promote the formation of foam cells and contribute to plaque accumulation in arteries during atherogenesis. In addition, activation of LOX-1, TLR2, TLR4, RAGE, and AT1R by their specific agonists in SMCs can contribute to the production of ROS and pro-inflammatory cytokines via NF-κB-signaling to further upregulate inflammation, endothelial dysfunction, and macrophage recruitment. Taurine and terpenes modulate atherogenesis by suppressing inflammation through inhibiting NF-κB-induced pro-inflammatory cytokine production signaling pathways, reducing ROS and HMGB1, and promoting hypocholesterolemia and reduced LDL plasma concentration. Additionally, both taurine and terpenes can reduce hypertension by inhibiting L-type Ca^2+^ channels and preventing subsequent Ca^2+^ influx to prevent vasoconstrictive responses. *Abbreviations:* AngII, angiotensin II; AT_1_R, angiotensin type 1-receptor; CB_2_R, cannabinoid type 2-receptor; EC, endothelial cell; HHcy, hyperhomocysteinemia; HMGB-1, high-mobility group box-1; HOCl, hypochlorous acid; ICAM-1, intracellular adhesion molecule-1; IL, interleukin; LDL, low-density lipoprotein; LOX-1, lectin-like oxidized low-density lipoprotein receptor-1; MCP-1, monocyte chemoattractant protein-1; NF-κB, nuclear factor kappa-light-chain-enhancer of activated B cells; NOX, nicotinamide adenine dinucleotide phosphate oxidase; oxLDL, oxidized low-density lipoprotein; ROS, reactive oxygen species; RAGE, receptor for advanced glycation end-products; SMC, smooth muscle cell; TLR, toll-like receptor; TNF-*α*, tumor necrosis factor alpha; VCAM-1, vascular cell adhesion molecule-1. Figure designed with Biorender.com.

## Data Availability

Data sharing not applicable.

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
