# Peer review of "Combination of Taurine and Black Pepper Extract as a Treatment for Cardiovascular and Coronary Artery Diseases"

_nutrients, 2023, doi:10.3390/nu15112562_

Round 1
Reviewer 1 Report
Authors reviewed the potential role of taurine and black pepper in CV diseases, as inducers of anti -inflammatory and -oxidative actions which reduce evolution of atherosclerosis phenomena. The review is well written and designed, proposing interesting areas of research and more targets for therapy. However, some issues should be corrected before being published.
- Firstly, the review can be too dense. The anti-oxidant capacity of terpens, which is more known by readers, could be reduced
- Following evidence you showed, the anti-inflammatory proprieties of taurine and black pepper could be more systemic than specific for CV system. Please, discuss
- What is the potential influence of these terpens on the short chain fatty acid production?
- How can be these molecules affected in dysbiosis? Could bacteria modify terpens production/metabolism?
- The blockade of the non-canonical NFkB pathway by terpens must be included as a scheme. Please, also revise some redundancies in the schemes
- Could terpens attenuate inflammasome activation? This mechanism may be of highly interest in the CV system
- Terpens might reduce ROS production from mitochondria. However, what could be the side effects of lessening mitochondrial ETC for the CV system? Cardiac cells, mainly cardiomyocytes, use mitochondrial ATP as a main source of energy.
Author Response
Reviewer 1#
Authors reviewed the potential role of taurine and black pepper in CV diseases, as inducers of anti -inflammatory and -oxidative actions which reduce evolution of atherosclerosis phenomena. The review is well written and designed, proposing interesting areas of research and more targets for therapy. However, some issues should be corrected before being published.
Comments:
Comment 1: Firstly, the review can be too dense. The anti-oxidant capacity of terpens, which is more known by readers, could be reduced
Thank you for your valuable feedback on our manuscript. We have considered your suggestion that the antioxidant capacity of taurine and terpenes can be too dense and agree. We have revised section 3 “Anti-oxidative effects of taurine and terpene” to improve the overall readability of the review.
Comment 2: Following evidence you showed, the anti-inflammatory proprieties of taurine and black pepper could be more systemic than specific for CV system. Please, discuss
We appreciate this comment on the systemic anti-inflammatory effective of taurine/terpenes. To address this, we have included:
Line 108-110: as well as other conditions such as inflammatory bowel disease [47], rheumatoid arthritics [47], and traumatic brain injury where inflammation is a well-known critical event [45].
Line 149-152: Pre-treatment of limonene (25, 50 and 75 mg/kg) inhibited release of prom-inflammatory cytokines (i.e., TNF-α, IL-1β and IL-6) and prevented phosphorylation of p38-MAPK, IκBα, NF-κB p65, JNK and ERK in mice with lipopolysaccharide (LPS)-induced acute lung injury [63].
Line 178-180: Recently, NLRP3 has been shown to play an indispensable role in the development of vascular diseases, including hypertension, atherosclerosis, and CAD, as well as diabetes and neurological disorders [77].
Comment 2: What is the potential influence of these terpenes on the short chain fatty acid production? How can be these molecules affected in dysbiosis? Could bacteria modify terpenes production/metabolism?
Thank you for this comment regarding effects on short chain fatty acid production and effects of compounds on dysbiosis. We now included the following:
Line 713-728: Furthermore, imbalances in the gut microbiome have been linked to the pathogenesis of atherosclerosis [275,276]. Short-chain fatty acids produced by gut bacterial fermentation contain a beneficial effect against systemic inflammation, inhibiting pro-inflammatory cytokines and promoting the preservation of endothelial function, thereby exhibiting anti-atherosclerotic actions [277,278]. Imbalances in gut microbiota, resulting in pathological bacteria can decrease the availability of short chain fatty acids and induce system inflammation to aggravate plaque development in atherosclerosis [279,280]. Taurine and terpenes are both known to contain antimicrobial activities [281]. In healthy mice, taurine treatment (165 mg/kg) inhibits growth of harmful bacteria and increases the production of short chain fatty acids [282]. Major black pepper terpene limonene has been shown to positively modulate gut microbiota in high-fat fed animal models [283], while a meta-analysis on the antibacterial actions of α-pinene reported that the terpenoid provides protective effects against a wide variety of pathogenic gut bacterial, including E. coli [284]. These studies suggest that a potential aspect of taurine and terpenes anti-atherosclerotic actions may occur through modulation of gut microbiome and potential increase in short chain fatty acids.
.
Comment 4: The blockade of the non-canonical NFkB pathway by terpenes must be included as a scheme. Please, also revise some redundancies in the schemes.
Thank you for this feedback on our manuscript. We have now revised the NF-kB pathway and decided to omit the non-canonical pathway as the effects of taurine and terpenes are not relevant to that scheme.
Comment 5: Could terpenes attenuate inflammasome activation? This mechanism may be of highly interest in the CV system.
Thank you for bringing this to our attention, we also believe that showing the effects of compounds on inhibiting inflammasome activation is warranted. We have now added:
Line 175-193: NLR family pyrin domain-containing 3 (NLRP3) inflammasome is a critical component of the innate immune system, responsible for the production of pro-inflammatory cytokines in response to cellular damage [76]. Recently, NLRP3 has been shown to play an indispensable role in the development of vascular diseases, including hypertension, atherosclerosis, and CHD, as well as diabetes and neurological disorders [77]. Increasing evidence supports NLRP3 as a new target to inhibit inflammatory diseases [78]. NLRP3 is regulated by NF-κB signaling pathways, therefore, the ability of taurine and terpenes to inhibit NF-κB as we have perilously highlighted may be a pathway for preventing NLRP3 activation. Although taurine has been shown to attenuate in vitro and ex vivo inflammation through inhibition of NF-κB-dependent NLRP3 activation [79,80], studies on isolated effect of black pepper terpenes on inflammasome inhibition are limited. A study conducted on the essential oil of pomelo peel, in which limonene (55.92%) was the most abundant compound significantly inhibited inflammation mediated by NLRP3 activation in rats [81]. Additionally, Makauy leaf ethanol extract consisting of limonene (5.6%) and BCP (4.9%) inhibited the in vitro expression of NLRP3 in J774A.1 mouse macrophage cells stimulated with LPS-induced inflammation [82]. Lastly, CB2R agonist JWH-133 has been shown to produce cardioprotective effects against MI in rats by suppressing NLPR3-releated mechanisms suggesting that BCP, a CB2R agonist may also suppress inflammation through NLPR2 inhibition [83].
Comment 6: Terpenes might reduce ROS production from mitochondria. However, what could be the side effects of lessening mitochondrial ETC for the CV system? Cardiac cells, mainly cardiomyocytes, use mitochondrial ATP as a main source of energy.
Thank you for sharing this concern. To address this, we have provided more clarity on the protective role of taurine on restoring mitochondrial ETC function by preventing overproduction of ROS that occurs during dysfunction, which reads as follows:
Line338-347: An increase in intracellular Ca2+ can impair mitochondrial membrane depolarization, causing a depletion in energy production. Additionally, increased ROS production during mitochondrial dysfunction is associated with decreased expression of mitochondrial ND5 and ND6 subunits of complex I of the electron transport chain (ETC), which can cause the ETC to become sluggish, creating a bottleneck and increasing diversion of electrons to oxygen to from superoxide (O2-) free radical [129]. Taurine is a component of mitochondrial tRNA [130], therefore, it is suggested that the association between taurine and tRNA acts to regulate mitochondrial protein synthesis, thereby enhancing ECT activity, improving energy metabolism and protecting the mitochondria against excessive O2- generation [126,131].

Reviewer 2 Report
1. End each paragraph with a two-three lines conclusion
2. Include figures summarizing the main findings and depicting the mode of action
3. Try to concentrate your review article by discussing the articles published in the last four years
4. Try to separate clearly in your review article clinical results from experimental studies (i.e. studies on cells, in vitro, animals). For clinical studies, include systematic reviews/meta-analyses, if available.
5. There is numerous spelling, grammatical and typographical errors throughout the manuscript the authors are requested to check the entire manuscript for errors.
6. Chemical structures must be drawn using same styles and are not of the same size
7. Though there is a paragraph dedicated to future perspectives it will be better if the author could draft a proper conclusion for the paper.
Minor editing of English language required - please, revise whole text.
Author Response
Reviewer #2
Comments:
Comment 1: End each paragraph with two-three lines conclusion.
We appreciate this comment and have added conclusions for each of our paragraphs.
Comment 2: Include figures summarizing the main findings and depicting the mode of action
Thank you for this comment. We have provided figures addressing the main issues of CVDs and showing the potential modes of action.
Comments 3: Try to concentrate your review article by discussing the articles published in the last four years.
This comment is highly appreciated. To address this concern, we have concentrated to reviews published recently where possible.
Comments 4: Try to separate clearly in your review article clinical results from experimental studies (i.e. studies on cells, in vitro, animals). For clinical studies, include systematic reviews/meta-analyses, if available.
Thank you for this feedback on our manuscript. In our review article, we have provided additional information on type of study as you suggested.
Comments 5: There are numerous spelling, grammatical and typographical errors throughout the manuscript; the authors are requested to check the entire manuscript for errors.
Thank you for bringing this to our attention. We checked and fixed the manuscript for errors.
Comments 6: Chemical structures must be drawn using the same styles and are not of the same size.
We appreciate bringing this to our attention. We have now re-sized chemical structures.
Comments 7: Though there is a paragraph dedicated to future perspectives it will be better if the author could draft a proper conclusion for the paper.
This feedback is greatly appreciated. We have revised our conclusion which now reads as follows:
Line 771-791: Herein, this comprehensive literature review provides a plethora of evidence supporting the use of taurine and the terpene constituents (i.e., β-caryophyllene; É‘-pinene; β-pinene; É‘-humulene; limonene; and sabinene) that make up PhytoCann BP® black pepper extract [277], to target risk factors for CVDs (i.e., hypertension and hyperhomocysteinemia) by promoting anti-inflammatory effects through suppression of TLR/NF-κB/MAPK-dependent proinflammatory signaling pathways; anti-oxidative effects via restoring mitochondrial health, preventing the production of oxidative free radicals, and promoting the scavenging of free radicals; anti-atherogenic effect associated with the suppression of hypercholesterolemia, hyperlipidemia, and inhibition of cellular adhesion molecules to prevent atherogenic plaque development. These mechanisms of taurine and terpenes could potentially combine, to act in synergy to become a novel treatment for CVDs, including CAD, HF, MI, and atherosclerotic disease. Importantly, we also show unique cardioprotective abilities of taurine and black pepper extract. For example, taurine appears to be able to target the RAS while black pepper extract is able to quench HMGB-1 release, and thus providing the theoretical basis for the development of a combined treatment. Based on previous dosages used in the literature and clinical studies, we propose a theoretical dose of 500 mg – 3 g of taurine [115,173,176,177,224] and 5 – 150 mg of black pepper extract [69,105,162] to be taken orally in capsule form, twice a day. Further clinical studies are required to confirm this hypothesis that a combined taurine and black pepper supplementation may be a viable adjunct treatment of CVD and its complications in humans.
Minor editing of English language required - please, revise whole text.
Thank you for bringing this to our attention. We have read and corrected spelling and grammar errors.
Reviewer 3 Report
This comprehensive review of the literature focuses on determining whether the combination of taurine and black pepper extract is an appealing natural treatment for reducing cardiovascular diseases risk factors (i.e., hypertension and hyperhomocysteinemia) and driving anti-inflammatory, antioxidative and anti-atherosclerotic mechanisms during coronary artery disease, heart failure, myocardial infarction, and atherosclerotic disease. The topic is interesting for treating ardiovascular and coronary artery diseases. However, there are some problems in the review. Since the aim of the review is to review the combined effects of taurine and black pepper extract as a treatment for cardiovascular and coronary artery diseases, it is necessary to introduce the combined effects of taurine and black pepper extract instead of the respective effect. Moreover, to have the best treatment effects, the proportion and composition and the administration method of the main components in taurine and black pepper are also needed to provide. The most important, it is necessary to provide the background and theoretical basis for the combined effects of taurine and black pepper, and explain why they can be used together as a treatment for cardiovascular and coronary artery diseases.
Minor concerns:
1. The keywords are inappropriate.
2. Some doi numbers in references are missing.
Minor editing of English language required
Author Response
Reviewer #3
This comprehensive review of the literature focuses on determining whether the combination of taurine and black pepper extract is an appealing natural treatment for reducing cardiovascular diseases risk factors (i.e., hypertension and hyperhomocysteinemia) and driving anti-inflammatory, antioxidative and anti-atherosclerotic mechanisms during coronary artery disease, heart failure, myocardial infarction, and atherosclerotic disease. The topic is interesting for treating cardiovascular and coronary artery diseases. However, there are some problems in the review.
Comment 1. Since the aim of the review is to review the combined effects of taurine and black pepper extract as a treatment for cardiovascular and coronary artery diseases, it is necessary to introduce the combined effects of taurine and black pepper extract instead of the respective effect.
We appreciate your feedback and the opportunity to improve our work. To address your concern, we have introduced the potential synergistic effects that taurine and terpenes may provide through their effects on different mechanism to provided therapeutic treatment of cardiovascular aetiologies.
Comment 2. Moreover, to have the best treatment effects, the proportion and composition and the administration method of the main components in taurine and black pepper are also needed to provide. The most important, it is necessary to provide the background and theoretical basis for the combined effects of taurine and black pepper and explain why they can be used together as a treatment for cardiovascular and coronary artery diseases.
Thank you for bringing this to our attention. We agree that it is needed for us to give a theoretical dose and mode of supplement. We have now added this to our conclusion.
Minor concerns:
- The keywords are inappropriate.
Thank you for this comment, we have since removed majority of keywords to better suit the review.
- Some doi numbers in references are missing.
Thank you for bringing to our attention. We have revised articles missing doi numbers; however, we were unable to locate the DOIs for the following papers:
- Kim, B.S.; Cho, I.S.; Park, S.Y.; Schuller-Levis, G.; Levis, W.; Park, E. Taurine chloramine inhibits NO and TNF-alpha production in zymosan plus interferon-gamma activated RAW 264.7 cells. J Drugs Dermatol 2011, 10, 659-665.
- Santiago, J.V.A.; Jayachitra, J.; Shenbagam, M.; Nalini, N. d-limonene attenuates blood pressure and improves the lipid and antioxidant status in high fat diet and L-NAME treated rats. Journal of Pharmaceutical Sciences and Research 2010, 2, 752
- Park, E.; Schuller-Levis, G.; Quinn, M.R. Taurine chloramine inhibits production of nitric oxide and TNF-alpha in activated RAW 264.7 cells by mechanisms that involve transcriptional and translational events. J Immunol 1995, 154, 4778-4784.
Round 2
Reviewer 3 Report
The authors have revised the manuscript according to the reviewer's comment.